# On the Role of Sparsity and DAG Constraints for Learning Linear DAGs

**Ignavier Ng**
University of Toronto

**AmirEmad Ghassami**
Johns Hopkins University

**Kun Zhang**
Carnegie Mellon University

## Abstract

Learning graphical structures based on Directed Acyclic Graphs (DAGs) is a challenging problem, partly owing to the large search space of possible graphs. A recent line of work formulates the structure learning problem as a continuous constrained optimization task using the least squares objective and an algebraic characterization of DAGs. However, the formulation requires a hard DAG constraint and may lead to optimization difficulties. In this paper, we study the asymptotic role of the sparsity and DAG constraints for learning DAG models in the linear Gaussian and non-Gaussian cases, and investigate their usefulness in the finite sample regime. Based on the theoretical results, we formulate a likelihood-based score function, and show that one only has to apply soft sparsity and DAG constraints to learn a DAG equivalent to the ground truth DAG. This leads to an unconstrained optimization problem that is much easier to solve. Using gradient-based optimization and GPU acceleration, our procedure can easily handle thousands of nodes while retaining a high accuracy. Extensive experiments validate the effectiveness of our proposed method and show that the DAG-penalized likelihood objective is indeed favorable over the least squares one with the hard DAG constraint.

## 1  Introduction

Learning graphical structures from data based on Directed Acyclic Graphs (DAGs) is a fundamental problem in machine learning, with applications in many areas such as biology [40] and healthcare [26]. It is clear that the learned graphical models may not be interpreted causally [33, 48]. However, they provide a compact, yet flexible, way to decompose the joint distribution. Under further conditions, these graphical models may have causal interpretations or be converted to representations (e.g., Markov equivalence classes) that have causal interpretations.

Two major classes of structure learning methods are constraint- and score-based methods. Constraint-based methods, including PC [46] and FCI [47, 11], utilize conditional independence tests to recover the Markov equivalence class under faithfulness assumption. On the other hand, score-based methods formulate the problem as optimizing a certain score function [19, 10, 49, 45]. Due to the large search space of possible graphs [9], most score-based methods rely on local heuristics, such as GES [10].

Recently, Zheng et al. [54] have introduced the NOTEARS method which formulates the structure learning problem as a continuous constrained optimization task, leveraging an algebraic characterization of DAGs. NOTEARS is specifically developed for linear DAGs, and has been extended to handle nonlinear cases via neural networks [21, 53, 28, 29, 25, 55]. Other related works include DYNOTEARS [32] that focuses on time-series data, and [56] that uses reinforcement learning to find the optimal DAGs. NOTEARS and most of its extensions adopt the least squares objective, which is related to but does not directly maximize the data likelihood. Furthermore, their formulations require a hard DAG constraint which may lead to optimization difficulties (see Section 2.2).

In this work, we investigate whether such a hard DAG constraint and another widely used sparsity constraint are necessary for learning DAGs. Inspired by that, we develop a likelihood-based structure

learning method with continuous unconstrained optimization, called *Gradient-based Optimization of dag-penalized Likelihood for learning linEar dag Models* (GOLEM). Our contributions are:

- We compare the differences between the regression-based and likelihood-based objectives for learning linear DAGs.
- We study the asymptotic role of the sparsity and DAG constraints in the general linear Gaussian case and other specific cases including linear non-Gaussian model and linear Gaussian model with equal noise variances. We also investigate their usefulness in the finite sample regime.
- Based on the theoretical results, we formulate a likelihood-based score function, and show that one only has to apply soft sparsity and DAG constraints to learn a DAG equivalent to the ground truth DAG. This removes the need for a hard DAG constraint[1][54] and leads to an unconstrained optimization problem that is much easier to solve.
- We demonstrate the effectiveness of our DAG-penalized likelihood objective through extensive experiments and an analysis on the bivariate linear Gaussian model.

The rest of the paper is organized as follows: We review the linear DAG model and NOTEARS in Section 2. We then discuss the role of sparsity and DAG constraints under different settings in Section 3. Based on the theoretical study, we formulate a likelihood-based method in Section 4, and compare it to NOTEARS and the least squares objective. The experiments in Section 5 verify our theoretical study and the effectiveness of our method. Finally, we conclude our work in Section 6.

## 2  Background

### 2.1  Linear DAG Model

A *DAG model* defined on a set of random variables $X = (X_1, \ldots, X_d)$ consists of (1) a DAG $G = (V(G), E(G))$ that encodes a set of conditional independence assertions among the variables, and (2) the joint distribution $P(X)$ (with density $p(x)$) that is Markov w.r.t. the DAG $G$, which factors as $p(x) = \prod_{i=1}^{d} p(x_i | x_{\mathsf{PA}_i^G})$, where $\mathsf{PA}_i^G = \{j \in V(G) : X_j \to X_i \in E(G)\}$ denotes the set of parents of $X_i$ in $G$. Under further conditions, these edges may have causal interpretations [33, 48]. In this work, we focus on the *linear DAG model* that can be equivalently represented by a set of linear Structural Equation Models (SEMs), in which each variable obeys the model $X_i = B_i^{\mathsf{T}} X + N_i$, where $B_i$ is a coefficient vector and $N_i$ is the exogenous noise variable corresponding to variable $X_i$. In matrix form, the linear DAG model reads $X = B^{\mathsf{T}} X + N$, where $B = [B_1 | \cdots | B_d]$ is a weighted adjacency matrix and $N = (N_1, \ldots, N_d)$ is a noise vector with independent elements. The structure of $G$ is defined by the nonzero coefficients in $B$, i.e., $X_j \to X_i \in E(G)$ if and only if the coefficient in $B_i$ corresponding to $X_j$ is nonzero. Given i.i.d. samples $\mathbf{x} = \left\{x^{(k)}\right\}_{k=1}^{n}$ from the distribution $P(X)$, our goal is to infer the matrix $B$, from which we may recover the DAG $G$ (or vice versa).[2]

### 2.2  The NOTEARS Method

Recently, Zheng et al. [54] have proposed NOTEARS that formulates the problem of learning linear DAGs as a continuous optimization task, leveraging an algebraic characterization of DAGs via the trace exponential function. It adopts a *regression-based* objective, i.e., the *least squares* loss, with $\ell_1$ penalty and a hard DAG constraint. The constrained optimization problem is then solved using the augmented Lagrangian method [5], followed by a thresholding step on the estimated edge weights.

In practice, the hard DAG constraint requires careful fine-tuning on the augmented Lagrangian parameters [6, 7, 31]. It may also encounter numerical difficulties and ill-conditioning issues as the penalty coefficient has to go to infinity to enforce acyclicity, as demonstrated empirically by Ng et al. [30]. Moreover, minimizing the least squares objective is related to but does not directly maximize the data likelihood because it does not take into account the log-determinant (LogDet) term of the likelihood (see Section 4.4). By contrast, we develop a *likelihood-based* method that directly maximizes the data likelihood, and requires only soft sparsity and DAG constraints.

# 3 Asymptotic Role of Sparsity and DAG Constraints

In this section we study the asymptotic role of the sparsity and DAG constraints for learning linear DAG models. Specifically, we aim to investigate with different model classes, whether one should consider the DAG constraint as a hard or soft one, and what exactly one benefits from the sparsity constraint. We consider a score-based structure learning procedure that optimizes the following score function w.r.t. the weighted adjacency matrix $B$ representing a directed graph:

$$\mathcal{S}(B; \mathbf{x}) = \mathcal{L}(B; \mathbf{x}) + R_{sparse}(B) + R_{DAG}(B), \qquad (1)$$

where $\mathcal{L}(B; \mathbf{x})$ is the Maximum Likelihood Estimator (MLE), $R_{sparse}(B)$ is a penalty term encouraging sparsity, i.e., having fewer edges, and $R_{DAG}(B)$ is a penalty term encouraging DAGness on $B$. The penalty coefficients can be selected via cross-validation in practice.

It is worth noting that the sparsity (or frugality) constraint has been exploited to find the DAG or its Markov equivalence class with as few edges as possible, searching in the space of DAGs or equivalence classes. In particular, permutation-based methods have been developed to find the sparsest DAG across possible permutations of the variables [45, 38]. This type of methods may benefit from smart optimization procedures, but inevitably they involve combinatorial optimization. Different from previous work, in this paper we do not necessarily constrain the search space to be acyclic in a hard manner, but the estimated graph will be a DAG if the ground truth is acyclic.

We will describe our specific choices of the penalty functions in Section 4. Throughout the paper, we assume that the ground truth structure is a DAG. We are concerned with two different cases. One is the general linear Gaussian model (i.e., assuming nonequal noise variances), for which it is known that the underlying DAG structure is not identifiable from the data distribution only [24]. In the other case, the underlying DAG model is asymptotically identifiable from the data distribution itself, with or without constraining the search space to be the class of DAGs.

## 3.1 General Linear Gaussian Case

We first study the specific class of structures for which the sparsity penalty term $R_{sparse}(B)$ is sufficient for the MLE to asymptotically learn a DAG equivalent to the ground truth DAG, i.e., the DAG penalty term $R_{DAG}(B)$ is not needed. Then we show that for the general structures, adding $R_{DAG}(B)$ guarantees learning a DAG equivalent to the ground truth DAG. We first require a notion of equivalence to be able to investigate the consistency of the approach.

Ghassami et al. [16] have introduced a notion of equivalence among directed graphs, called quasi equivalence, as follows: For a directed graph $G$, define the distribution set of $G$, denoted by $\Theta(G)$, as the set of all precision matrices (equivalently, distributions) that can be generated by $G$ for different choices of exogenous noise variances and edge weights in $G$. Define a *distributional constraint* as any equality or inequality constraint imposed by $G$ on the entries of precision matrix $\Theta$. Also, define a *hard constraint* as a distributional constraint for which the set of the values satisfying that constraint is Lebesgue measure zero over the space of the parameters involved in the constraint. The set of hard constraints of a directed graph $G$ is denoted by $H(G)$. Note that the notion of hard constraint here is different from the hard DAG constraint used by Zheng et al. [54].

**Definition 1 (Quasi Equivalence).** *Let $\theta_G$ be the set of linearly independent parameters needed to parameterize any distribution $\Theta \in \Theta(G)$. For two directed graphs $G_1$ and $G_2$, let $\mu$ be the Lebesgue measure defined over $\theta_{G_1} \cup \theta_{G_2}$. $G_1$ and $G_2$ are quasi equivalent if $\mu(\theta_{G_1} \cap \theta_{G_2}) \neq 0$.*

Roughly speaking, two directed graphs are quasi equivalent if the set of distributions that they can both generate has a nonzero Lebesgue measure. See Appendix A for an example. Definition 1 implies that if directed graphs $G_1$ and $G_2$ are quasi equivalent, they share the same hard constraints.

The following two assumptions are required for the task of structure learning from observational data.

**Assumption 1 (G-faithfulness Assumption).** *A distribution $\Theta$ is generalized faithful (g-faithful) to structure $G$ if $\Theta$ satisfies a hard constraint $\kappa$ if and only if $\kappa \in H(G)$. We say that the g-faithfulness assumption is satisfied if the generated distribution is g-faithful to the ground truth structure.*

**Assumption 2.** *Let $E(G)$ be the set of edges of $G$. For a DAG $G^*$ and a directed graph $\hat{G}$, we have the following statements.*
*(a) If $|E(\hat{G})| \leq |E(G^*)|$, then $H(\hat{G}) \not\subset H(G^*)$.*
*(b) If $|E(\hat{G})| < |E(G^*)|$, then $H(\hat{G}) \not\subseteq H(G^*)$.*

Assumption 1 is an extension of the well-known faithfulness assumption [48]. The intuition behind Assumption 2 is that in DAGs all the parameters can be chosen independently. Hence, each parameter introduces an independent dimension to the distribution space. Therefore, if a DAG and another directed graph $\hat{G}$ have the same number of edges, then the distribution space of the DAG cannot be a strict subset with lower dimension of the distribution space of $\hat{G}$. Note that the assumption holds if $\hat{G}$ is also a DAG. Ghassami et al. [16] have showed that the g-faithfulness assumption is a mild one in the sense that the Lebesgue measure of the distributions not g-faithful to the ground truth is zero, and showed that under Assumption 1 and a condition similar to Assumption 2, the underlying directed graph can be identified up to quasi equivalence and proposed an algorithm to do so.

In the following, we first consider the case that the DAG penalty term $R_{DAG}(B)$ in expression (1) is not needed. The following condition is required for this case.

**Assumption 3 (Triangle Assumption).** *A DAG satisfies the triangle assumption if it does not have any triangles (i.e., 3-cycles) in its skeleton.*

As an example, any polytree satisfies the triangle assumption.

**Theorem 1.** *If the underlying DAG satisfies Assumptions 1-3, a sparsity penalized MLE asymptotically returns a DAG quasi equivalent to the ground truth DAG.*

If we relax the triangle assumption, the global minimizer of $\mathcal{L}(B; \mathbf{x}) + R_{sparse}(B)$ can be cyclic. However, the following theorem shows that even in this case, some global minimizers are still acyclic.

**Theorem 2.** *If the underlying DAG satisfies Assumptions 1 and 2, the output of sparsity penalized MLE asymptotically has the same number of edges as the ground truth.*

This motivates us to add the DAG penalty term $R_{DAG}(B)$ to the score function (1) to prefer a DAG solution to a cyclic one with the same number of edges.

**Corollary 1.** *If the underlying DAG satisfies Assumptions 1 and 2, a sparsity and DAG penalized MLE asymptotically returns a DAG quasi equivalent to the ground truth DAG.*

The proofs of Theorems 1 and 2 are given in Appendix B. Corollary 1 implies that, under mild assumptions, one only has to apply soft sparsity and DAG constraints to the likelihood-based objective instead of constraining the search space to be acyclic in a hard manner, and the estimated graph will be a DAG up to quasi equivalence.

## 3.2 With Identifiable Linear DAG Models

In a different line of research, linear DAG models may be identifiable under specific assumptions. Suppose that the ground truth is a DAG. There are two types of identifiability results for the underlying DAG structure. One does not require the constraint that the search space is the class of DAGs; a typical example is the Linear Non-Gaussian Acyclic Model (LiNGAM) [43], where at most one of the noise terms follows Gaussian distribution. In this case, it has been shown that as the sample size goes to infinity, among all directed graphical models that are acyclic or cyclic, only the underlying graphical model, which is a DAG, can generate exactly the given data distribution, thanks to the identifiability results of the Independent Component Analysis (ICA) problem [20]. Hence, asymptotically speaking, given observational data generated by the LiNGAM, we do not need to enforce the sparsity or DAG constraint in the estimation procedure that maximizes the data likelihood, and the estimated graphical model will converge to the ground truth DAG. However, on finite samples, one still benefits from enforcing the sparsity and DAG constraints by incorporating the corresponding penalty term: because of random estimation errors, the linear coefficients whose true values are zero may have nonzero estimated values in the maximum likelihood estimate, and the constraints help set them to zero.

By contrast, the other type of identifiable linear DAG model constrains the estimated graph to be in the class of DAGs. An example is the linear Gaussian model with equal noise variances [34]. In the proof of the identifiability result [34, Theorem 1], it shows that when the sample size goes to infinity, there is no other DAG structure that can generate the same distribution. In theory, it is unclear whether any cyclic graph is able to generate the same distribution; however, we strongly believe that in this identifiability result, one has to apply the sparsity or DAG constraint, as suggested by our empirical results (see Section 5.1) and an analysis in the bivariate case (Proposition 1).

Note that whether one benefits from the above identifiability results depends on the form of likelihood function. If it does not take into account the additional assumptions that give rise to identifiability and relies on the general linear Gaussian model, then the analysis in Section 3.1 still applies.

# 4 GOLEM: A Continuous Likelihood-Based Method

The theoretical results in Section 3 suggest that likelihood-based objective with soft sparsity and DAG constraints asymptotically returns a DAG equivalent to the ground truth DAG, under mild assumptions. In this section, we formulate a continuous likelihood-based method based on these constraints, and describe the post-processing step and computational complexity. We then compare the resulting method to NOTEARS and the least squares objective.

## 4.1 Maximum Likelihood Objectives with Soft Constraints

We formulate a score-based method to maximize the data likelihood of a linear Gaussian model, with a focus on continuous optimization. The joint distribution follows multivariate Gaussian distribution, which gives the following objective w.r.t. the weighted matrix $B$ representing a directed graph:

$$\mathcal{L}_1(B; \mathbf{x}) = \frac{1}{2} \sum_{i=1}^{d} \log \left( \sum_{k=1}^{n} \left( x_i^{(k)} - B_i^\mathsf{T} x^{(k)} \right)^2 \right) - \log |\det(I - B)|.$$

If one further assumes that the noise variances are equal (although they may be nonequal), it becomes

$$\mathcal{L}_2(B; \mathbf{x}) = \frac{d}{2} \log \left( \sum_{i=1}^{d} \sum_{k=1}^{n} \left( x_i^{(k)} - B_i^\mathsf{T} x^{(k)} \right)^2 \right) - \log |\det(I - B)|. \tag{2}$$

The objectives above are denoted as likelihood-NV and likelihood-EV, respectively, with derivations provided in Appendix C. Note that they give rise to the BIC score [42] (excluding complexity penalty term) assuming nonequal and equal noise variances, respectively, in the linear Gaussian setting.

In principle, one should use (a function of) the number of edges to assess the structure complexity, such as our study in Section 3.1 and the $\ell_0$ penalty from BIC score [10, 52, 38]. However, it is difficult to optimize such a score in practice (e.g., GES [10] adopts greedy search in the discrete space). To enable efficient continuous optimization, we use the $\ell_1$ penalty for approximation. Although the $\ell_1$ penalty has been widely used in regression tasks [50] to find sparse precision matrices [27, 15], it has been rarely used to directly penalize the likelihood function in the linear Gaussian case [3]. With soft $\ell_1$ and DAG constraints, the *unconstrained* optimization problems of our score functions are

$$\min_{B \in \mathbb{R}^{d \times d}} \quad \mathcal{S}_i(B; \mathbf{x}) = \mathcal{L}_i(B; \mathbf{x}) + \lambda_1 \|B\|_1 + \lambda_2 h(B), \tag{3}$$

where $i = 1, 2$, $\lambda_1$ and $\lambda_2$ are the penalty coefficients, $\|B\|_1$ is defined element-wise, and $h(B) = \mathrm{tr}\left(e^{B \circ B}\right) - d$ is the characterization of DAGness proposed by Zheng et al. [54]. It is possible to use the characterization suggested by Yu et al. [53], which is left for future work. The score functions $\mathcal{S}_i(B; \mathbf{x}), i = 1, 2$ correspond respectively to the likelihood-NV and likelihood-EV objectives with soft sparsity and DAG constraints, which are denoted as GOLEM-NV and GOLEM-EV, respectively.

Unlike NOTEARS [54] that requires a hard DAG constraint, we treat it as a soft one, and the estimated graph will (asymptotically) be a DAG if the ground truth is acyclic, under mild assumptions (cf. Section 3). This leads to the unconstrained optimization problems (3) that are much easier to solve. Detailed comparison of our proposed method to NOTEARS is further described in Section 4.4.

Similar to NOTEARS, the main advantage of the proposed score functions is that continuous optimization method can be applied to solve the minimization problems, such as first-order (e.g., gradient descent) or second-order (e.g., L-BFGS [8]) method. Here we adopt the first-order method Adam [23] implemented in `Tensorflow` [1] with GPU acceleration and automatic differentiation (see Appendix F for more details). Note, however, that the optimization problems inherit the difficulties of nonconvexity, indicating that they can only be solved to stationarity. Nonetheless, the empirical results in Section 5 demonstrate that this leads to competitive performance in practice.

**Initialization scheme.** In practice, the optimization problem of GOLEM-NV is susceptible to local solutions. To remedy this, we find that initializing it with the solution returned by GOLEM-EV dramatically helps avoid undesired solutions in our experiments.

## 4.2 Post-Processing

Asymptotically speaking, the estimated graph returned by GOLEM will, under mild assumptions, be acyclic (cf. Section 3). Nevertheless, due to finite samples and nonconvexity, the local solution obtained may contain several entries near zero and may not be exactly acyclic. We therefore set a small threshold $\omega$, as in [54], to remove edges with absolute weights smaller than $\omega$. The key idea is to "round" the numerical solution into a discrete graph, which also helps reduce false discoveries. If the thresholded graph contains cycles, we remove edges iteratively starting from the lowest absolute weights, until a DAG is obtained. In other words, one may gradually increase $\omega$ until the thresholded graph is acyclic. This heuristic is made possible by virtue of the DAG penalty term, since it pushes the cycle-inducing edges to small values.

## 4.3 Computational Complexity

Gradient-based optimization involves gradient evaluation in each iteration. The gradient of the LogDet term from the score functions $\mathcal{S}_i(B; \mathbf{x}), i = 1, 2$ is given by $\nabla_B \log |\det(I - B)| = -(I - B)^{-\mathsf{T}}$. This implies that $\mathcal{S}_i(W; \mathbf{x})$ and its gradient involve evaluating the LogDet and matrix inverse terms. Similar to the DAG penalty term with matrix exponential [2, 54], the $\mathcal{O}(d^3)$ algorithms of both these operations [51, 14] are readily available in multiple numerical computing frameworks [1, 18]. Our experiments in Section 5.4 demonstrate that the optimization could benefit from GPU acceleration, showing that the cubic evaluation costs are not a major concern.

## 4.4 Connection with NOTEARS and Least Squares Objective

It is instructive to compare the likelihood-EV objective (2) to the least squares, by rewriting (2) as

$$\mathcal{L}_2(B; \mathbf{x}) = \frac{d}{2} \log \ell(B; \mathbf{x}) - \log |\det(I - B)| + \frac{d}{2} \log 2n,$$

where $\ell(B; \mathbf{x}) = \frac{1}{2n} \sum_{i=1}^d \sum_{k=1}^n \left( x_i^{(k)} - B_i^{\mathsf{T}} x^{(k)} \right)^2$ is the least squares objective. One observes that the main difference lies in the LogDet term. Without the LogDet term, the least squares objective with $\ell_1$ penalty corresponds to a multiple-output lasso problem, which is decomposable into $d$ independent regression tasks. Thus, least squares objective tends to introduce cycles in the estimated graph (see Proposition 1 for an analysis in the bivariate case). Consider two variables $X_i$ and $X_j$ that are conditionally dependent given any subset of the remaining variables, then one of them is useful in predicting the other. That is, when minimizing the least squares for one of them, the other variable will tend to have a nonzero coefficient. If one considers those coefficients as weights of the graph, then the graph will have cycles. By contrast, the likelihood-based objective has an additional LogDet term that enforces a shared structure between the regression coefficients of different variables. We have the following lemma regarding the LogDet term, with a proof provided in Appendix D.

**Lemma 1.** *If a weighted matrix $B \in \mathbb{R}^{d \times d}$ represents a DAG, then*

$$\log |\det(I - B)| = 0.$$

Lemma 1 partly explains why a hard DAG constraint is needed by the least squares [54]: its global minimizer(s) is (are) identical to the likelihood-EV objective if the search space over $B$ is constrained to DAGs in a hard manner. However, the hard DAG constraint may lead to optimization difficulties (cf. Section 2.2). With a proper scoring criterion, the ground truth DAG should be its global minimizer, and thus the hard DAG constraint can be avoided. As suggested by our theoretical study, using likelihood-based objective, one may simply treat the constraint as a soft one, leading to an unconstrained optimization problem that is much easier to solve. In this case the estimated graph will be a DAG if the ground truth is acyclic, under mild assumptions. The experiments in Section 5 show that our proposed DAG-penalized likelihood objective yields better performance in most settings.

To illustrate our arguments above, we provide an example in the bivariate case. We consider the linear Gaussian model with ground truth DAG $G : X_1 \rightarrow X_2$ and equal noise variances, characterized by the following weighted adjacency matrix and noise covariance matrix:

$$B_0 = \begin{bmatrix} 0 & b_0 \\ 0 & 0 \end{bmatrix}, \Omega = \begin{bmatrix} \sigma^2 & 0 \\ 0 & \sigma^2 \end{bmatrix}, b_0 \neq 0. \tag{4}$$

We have the following proposition in the asymptotic case, with a proof given in Appendix E.

**Proposition 1.** *Suppose X follows a linear Gaussian model defined by Eq.* (4). *Then, asymptotically,*

*(a) $B_0$ is the unique global minimizer of least squares objective under a hard DAG constraint, but without the DAG constraint, the least squares objective returns a cyclic graph.*

*(b) $B_0$ is the unique global minimizer of likelihood-EV objective* (2) *under soft $\ell_1$ or DAG constraint.*

Therefore, without the DAG constraint, the least squares method never returns a DAG, while the likelihood objective does not favor cyclic over acyclic structures. This statement is also true in general: as long as a structure, be cyclic or acyclic, can generate the same distribution as the ground truth model, it can be the output of a likelihood score asymptotically.

Furthermore, Proposition 1 implies that both objectives produce asymptotically correct results in the bivariate case, under different conditions. Nevertheless, the condition required by the likelihood-EV objective (GOLEM-EV) is looser than that of the least squares (NOTEARS), as it requires only soft constraint to recover the underlying DAG instead of a hard one. This bivariate example serves as an illustration of our study in Section 3 which shows that GOLEM is consistent in the general case, indicating that the likelihood-based objective is favorable over the regression-based one.

## 5 Experiments

We first conduct experiments with increasing sample size to verify our theoretical study (Section 5.1). To validate the effectiveness of our proposed likelihood-based method, we compare it to several baselines in both identifiable (Section 5.2) and nonidentifiable (Section 5.3) cases. The baselines include FGS [37], PC [46, 36], DirectLiNGAM [44], NOTEARS-L1, and NOTEARS [54]. In Section 5.4, we conduct experiments on large graphs to investigate the scalability and efficiency of the proposed method. We then provide a sensitivity analysis in Section 5.5 to analyze the robustness of different methods. Lastly, we experiment with real data (Section 5.6). The implementation details of our procedure and the baselines are described in Appendices F and G.1, respectively.

Our setup is similar to [54]. The ground truth DAGs are generated from one of the two graph models, *Erdös–Rényi* (ER) or *Scale Free* (SF), with different graph sizes. We sample DAGs with $kd$ edges ($k = 1, 2, 4$) on average, denoted by ER$k$ or SF$k$. Unless otherwise stated, we construct the weighted matrix of each DAG by assigning uniformly random edge weights, and simulate $n = 1000$ samples based on the linear DAG model with different noise types. The estimated graphs are evaluated using normalized Structural Hamming Distance (SHD), Structural Intervention Distance (SID) [35], and True Positive Rate (TPR), averaged over 12 random simulations. We also report the normalized SHD computed over CPDAGs of estimated graphs and ground truths, denoted as SHD-C. Detailed explanation of the experiment setup and metrics can be found in Appendices G.2 and G.3, respectively.

### 5.1 Role of Sparsity and DAG Constraints

We investigate the role of $\ell_1$ and DAG constraints in both identifiable and nonidentifiable cases, by considering the linear Gaussian model with equal (*Gaussian-EV*) and nonequal (*Gaussian-NV*) noise variances, respectively. For *Gaussian-EV*, we experiment with the following variants: GOLEM-EV (with $\ell_1$ and DAG penalty), GOLEM-EV-L1 (with only $\ell_1$ penalty), and GOLEM-EV-Plain (without any penalty term), likewise for GOLEM-NV, GOLEM-NV-L1, and GOLEM-NV-Plain in the case of *Gaussian-NV*. Experiments are conducted on 100-node ER1 and ER4 graphs, each with sample sizes $n \in \{100, 300, 1000, 3000, 10000\}$. Here we apply only the thresholding step for post-processing.

Due to space limit, the results are shown in Appendix H.1. In the *Gaussian-EV* case, when the sample size is large, the graphs estimated by both GOLEM-EV and GOLEM-EV-L1 are close to the ground truth DAGs with high TPR, whereas GOLEM-EV-Plain has poor results without any penalty term. Notice also that the gap between GOLEM-EV-L1 and GOLEM-EV decreases with more samples, indicating that sparsity penalty appears to be sufficient to asymptotically recover the underlying DAGs. However, this is not the case for *Gaussian-NV*, as the performance of GOLEM-NV-L1 degrades without the DAG penalty term, especially for the TPR. These observations serve to corroborate our asymptotic study: (1) For the general *Gaussian-NV* case, although Theorem 1 states that sparsity penalty is sufficient to recover the underlying DAGs, the triangle assumption is not satisfied in this simulation. Corollary 1 has mild assumptions that apply here, implying that both sparsity and DAG penalty terms are required. (2) When the noise variances are assumed to be equal, i.e., in the *Gaussian-EV* case, Section 3.2 states that either sparsity or DAG penalty can help recover the ground

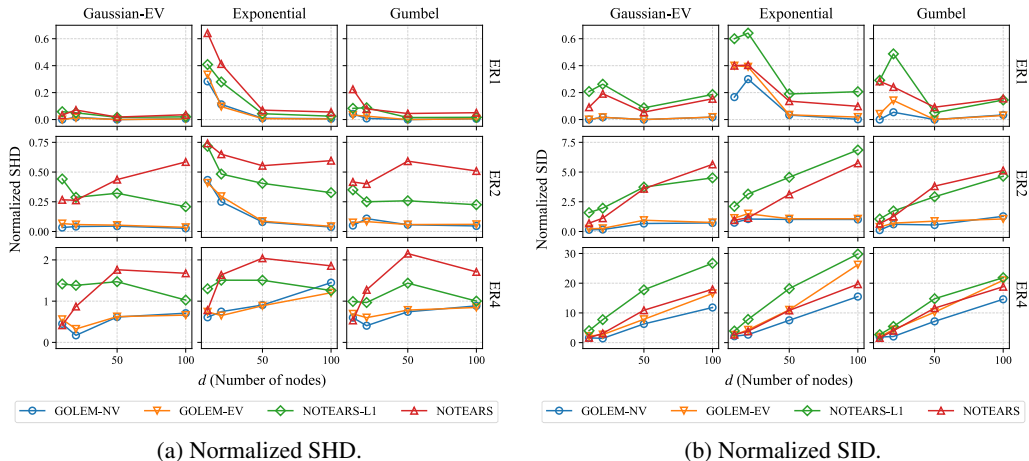

|   | (a) Normalized SHD. | (b) Normalized SID. |

Figure 1: Results in terms of normalized SHD and SID on ER graphs, with sample size $n = 1000$. Lower is better. Rows: ER$k$ denotes ER graphs with $kd$ edges on average. Columns: noise types. Since each panel has a number of lines, for better visualization we do not report the standard errors.

truth DAGs, thanks to the identifiability results. Nevertheless, DAG penalty is still very helpful in practice, especially on smaller sample sizes and denser graphs.

Since calculating the number of cycles in the estimated graphs may be too slow, we report the value of DAG penalty term $h(B)$ in Figures 4 and 5 as an indicator of DAGness. With $\ell_1$ and DAG penalty, the estimated graphs have low values of $h(B)$ across all settings. Consistent with our study, this implies that soft sparsity and DAG constraints are useful for both asymptotic cases and finite samples by returning solutions close to DAGs in practice, despite the nonconvexity of the optimization problem.

### 5.2 Numerical Results: Identifiable Cases

We examine the structure learning performance in the identifiable cases. In particular, we generate {ER1, ER2, ER4, SF4} graphs with different sizes $d \in \{10, 20, 50, 100\}$. The simulated data follows the linear DAG model with different noise types: *Gaussian-EV*, *Exponential*, and *Gumbel*.

For better visualization, the normalized SHD and SID of recent gradient-based methods (i.e., GOLEM-NV, GOLEM-EV, NOTEARS-L1, and NOTEARS) on ER graphs are reported in Figure 1, while complete results can be found in Appendix H.2. One first observes that gradient-based methods consistently outperform the other methods. Among gradient-based methods, GOLEM-NV and GOLEM-EV have the best performance in most settings, especially on large graphs. Surprisingly, these two methods perform well even in non-Gaussian cases, i.e., *Exponential* and *Gumbel* noise, although they are based on Gaussian likelihood. Also, we suspect that the number of edges whose directions cannot be determined is not high, giving rise to a high accuracy of our methods even in terms of SHD of the graphs. Consistent with previous work, FGS, PC, and DirectLiNGAM are competitive on sparse graphs (ER1), but their performance degrades as the edge density increases.

### 5.3 Numerical Results: Nonidentifiable Cases

We now conduct experiments in the nonidentifiable cases by considering the general linear Gaussian setting (i.e., *Gaussian-NV*) with {ER1, ER2, ER4} graphs. Hereafter we compare only with NOTEARS-L1, since it is the best performing baseline in the previous experiment.

Due to limited space, the results are given in Appendix H.3 with graph sizes $d \in \{10, 20, 50, 100\}$. Not surprisingly, GOLEM-NV shows significant improvement over GOLEM-EV in most settings, as the assumption of equal noise variances does not hold here. It also outperforms NOTEARS-L1 by a large margin on denser graphs, such as ER2 and ER4 graphs. Although GOLEM-EV and NOTEARS-L1 both assume equal noise variances (which do not hold here), it is interesting to observe that they excel in different settings: NOTEARS-L1 demonstrates outstanding performance on sparse graphs (ER1) but deteriorates on ER4 graphs, and vice versa for GOLEM-EV.

## 5.4 Scalability and Optimization Time

We compare the scalability of GOLEM-EV to NOTEARS-L1 using the linear DAG model with *Gaussian-EV* noise. We simulate $n = 5000$ samples on ER2 graphs with increasing sizes $d \in \{100, 200, 400, \ldots, 3200\}$. Due to the long optimization time, we are only able to scale NOTEARS-L1 up to 1600 nodes. The experiments of GOLEM-EV are computed on the P3 instance hosted on Amazon Web Services with a NVIDIA V100 GPU, while NOTEARS-L1 is benchmarked using the F4 instance on Microsoft Azure with four 2.4 GHz Intel Xeon CPU cores and 8 GB of memory.[3]

Here we report only the normalized SHD and TPR as the computation of SHD-C and SID may be too slow on large graphs. As depicted in Figure 8, GOLEM-EV remains competitive on large graphs, whereas the performance of NOTEARS-L1 degrades as the graph size increases, which may be ascribed to the optimization difficulties of the hard DAG constraint (cf. Section 2.2). The optimization time of GOLEM-EV is also much shorter (e.g., 12.4 hours on 3200-node graphs) owing to its parallelization on GPU, showing that the cubic evaluation costs are not a major concern. We believe that the optimization of NOTEARS-L1 could also be accelerated in a similar fashion.

## 5.5 Sensitivity Analysis of Weight Scale

We investigate the sensitivity to weight scaling as in [54]. We consider 50-node ER2 graphs with *Gaussian-EV* noise and edge weights sampled uniformly from $\alpha \cdot [-2, -0.5] \cup \alpha \cdot [0.5, 2]$ where $\alpha \in \{0.3, 0.4, \ldots, 1.0\}$. The threshold $\omega$ is set to 0.1 for all methods in this analysis.

The complete results are provided in Appendix H.5. One observes that GOLEM-EV has consistently low normalized SHD and SID, indicating that our method is robust to weight scaling. By contrast, the performance of NOTEARS-L1 is unstable across different weight scales: it has low TPR on small weight scales and high SHD on large ones. A possible reason for the low TPR is that the signal-to-noise ratio decreases when the weight scales are small, whereas for large ones, NOTEARS-L1 may have multiple false discoveries with intermediate edge weights, resulting in the high SHD.

## 5.6 Real Data

We also compare the proposed method to NOTEARS-L1 on a real dataset that measures the expression levels of proteins and phospholipids in human cells [40]. This dataset is commonly used in the literature of probabilistic graphical models, with experimental annotations accepted by the biological community. Based on $d = 11$ cell types and $n = 853$ observational samples, the ground truth structure given by Sachs et al. [40] contains 17 edges. On this dataset, GOLEM-NV achieves the best (unnormalized) SHD 14 with 11 estimated edges. NOTEARS-L1 is on par with GOLEM-NV with an SHD of 15 and 13 total edges, while GOLEM-EV estimates 21 edges with an SHD of 18.

## 6 Conclusion

We investigated whether the hard DAG constraint used by Zheng et al. [54] and another widely used sparsity constraint are necessary for learning linear DAGs. In particular, we studied the asymptotic role of the sparsity and DAG constraints in the general linear Gaussian case and other specific cases including the linear non-Gaussian model and linear Gaussian model with equal noise variances. We also investigated their usefulness in the finite sample regime. Our theoretical results suggest that when the optimization problem is formulated using the likelihood-based objective in place of least squares, one only has to apply soft sparsity and DAG constraints to asymptotically learn a DAG equivalent to the ground truth DAG, under mild assumptions. This removes the need for a hard DAG constraint and is easier to solve. Inspired by that, we developed a likelihood-based structure learning method with continuous unconstrained optimization, and demonstrated its effectiveness through extensive experiments in both identifiable and nonidentifiable cases. Using GPU acceleration, the resulting method can easily handle thousands of nodes while retaining a high accuracy. Future works include extending the current procedure to other score functions, such as BDe [19], decreasing the optimization time by deploying a proper early stopping criterion, devising a systematic way for thresholding, and studying the sensitivity of penalty coefficients in different settings.

## Broader Impact

The proposed method is able to estimate the graphical structure of a linear DAG model, and can be efficiently scaled up to thousands of nodes while retaining a high accuracy. DAG structure learning has been a fundamental problem in machine learning in the past decades, with applications in many areas such as biology [40]. Thus, we believe that our method could be applied for beneficial purposes.

Traditionally, score-based methods, such as GES [10], rely on local heuristics partly owing to the large search space of possible graphs. The formulation of continuous optimization for structure learning has changed the nature of the task, which enables the usage of well-studied gradient-based solvers and GPU acceleration, as demonstrated in Section 5.4.

Nevertheless, in practice, we comment that the graphical structures estimated by our method, as well as other structure learning methods, should be treated with care. In particular, they should be verified by domain experts before putting into decision-critical real world applications (e.g., healthcare). This is because the estimated structures may contain spurious edges, or may be affected by other factors, such as confounders, latent variables, measurement errors, and selection bias.

## Acknowledgments

The authors would like to thank Bryon Aragam, Shengyu Zhu, and the anonymous reviewers for helpful comments and suggestions. KZ would like to acknowledge the support by the United States Air Force under Contract No. FA8650-17-C-7715.

## Footnotes

[1]Using constrained optimization, the hard DAG constraint strictly enforces the estimated graph to be acyclic (up to numerical precision), which is stronger than a soft constraint.

[2]With a slight abuse of notation, we may also use $G$ and $B$ to refer to a directed graph (possibly cyclic) in the rest of the paper, depending on the context.

[3] For NOTEARS-L1, we have experimented with more CPU cores, such as the F16 instance on Azure with sixteen CPU cores and 32 GB of memory, but there is only minor improvement in the optimization time.

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
