[Supplementary Material]

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

# Appendices

## A  An Example of Quasi Equivalence

Figure 2: An example of quasi equivalence.

Here, we provide an example of two structures that are quasi equivalent to each other. Consider directed graphs $G_1$ and $G_2$ in Figure 2. Since $G_1$ is a complete DAG, it can generate any precision matrices. Consider an arbitrary precision matrix $\Theta$ generated by $G_1$. If $\Theta$ is representable by $G_2$, then we should be able to decompose it as $\Theta = QQ^\top$, where $Q$ has the following form.

$$Q = \begin{bmatrix} \sigma_1^{-1} & 0 & -\beta_{13}\sigma_3^{-1} \\ -\beta_{21}\sigma_1^{-1} & \sigma_2^{-1} & 0 \\ 0 & -\beta_{32}\sigma_2^{-1} & \sigma_3^{-1} \end{bmatrix}.$$

Therefore, it suffices to show that we have a matrix of form

$$\begin{bmatrix} a & 0 & b \\ c & d & 0 \\ 0 & e & f \end{bmatrix},$$

such that

$$a^2 + b^2 = \Theta_{11} \quad ac = \Theta_{12}$$
$$c^2 + d^2 = \Theta_{22} \quad bf = \Theta_{13}$$
$$e^2 + f^2 = \Theta_{33} \quad de = \Theta_{23}.$$

Then we have $\sigma_1 = a^{-1}$, $\sigma_2 = d^{-1}$, $\sigma_3 = f^{-1}$, $\beta_{13} = -b/f$, $\beta_{21} = -c/a$, $\beta_{32} = -e/d$.

Suppose that the value of $e$ is fixed. It should satisfy the following constraint:

$$e^2 = \Theta_{33} - \frac{\Theta_{13}^2}{\Theta_{11} - \frac{\Theta_{12}^2}{\Theta_{22} - \frac{\Theta_{23}^2}{e^2}}},$$

or equivalently,

$$(\Theta_{11}\Theta_{22} - \Theta_{12}^2)e^4 + (-\Theta_{11}\Theta_{22}\Theta_{33} - \Theta_{11}\Theta_{23}^2 + \Theta_{22}\Theta_{13}^2 + \Theta_{33}\Theta_{12}^2)e^2 + (\Theta_{11}\Theta_{33}\Theta_{23}^2 - \Theta_{13}^2\Theta_{23}^2) = 0,$$

which does not necessarily have a real root, and only for a non-measure zero subset of the distributions is satisfied.

## B  Proofs of Theorems 1 and 2

The following part is required for the proofs of both Theorems 1 and 2.

Let $G^*$ and $\Theta$ be the ground truth DAG and the generated distribution (precision matrix). Let $B$ and $\Omega$ be the weighted adjacency matrix and the diagonal matrix containing exogenous noise variances, respectively. Considering weights for penalty terms such that the likelihood term dominates asymptotically, we will find a pair $(\hat{B}, \hat{\Omega})$, such that $(I - \hat{B})\hat{\Omega}^{-1}(I - \hat{B})^\top = \Theta$ and denote the directed graph corresponding to $\hat{B}$ by $\hat{G}$. We have $\Theta \in \Theta(\hat{G})$, which implies that $\Theta$ contains all the distributional constraints of $\hat{G}$. Therefore, under the faithfulness assumption, we have $H(\hat{G}) \subseteq H(G^*)$. Due to the sparsity penalty we have $|E(\hat{G})| \leq |E(G^*)|$, otherwise the algorithm would have output $G^*$. By Assumption 2, we have $H(\hat{G}) \not\subset H(G^*)$. Now, from $H(\hat{G}) \subseteq H(G^*)$ and $H(\hat{G}) \not\subset H(G^*)$ we conclude that $H(\hat{G}) = H(G^*)$. Therefore, $\hat{G}$ is quasi equivalent to $G^*$.

**Proof of Theorem 1.**

To complete the proof of Theorem 1, we show that the output directed graph will be acyclic. We require the notion of virtual edge for the proof: For DAGs, under the Markov and faithfulness assumptions, a variable $X_i$ is adjacent to a variable $X_j$ if and only if $X_i$ and $X_j$ are dependent conditioned on any subset of the rest of the variables. This is not the case for cyclic directed graphs. Two nonadjacent variables $X_i$ and $X_j$ are dependent conditioned on any subset of the rest of the variables if they have a common child $X_k$ which is an ancestor of $X_i$ or $X_j$. In this case, we say that there exists a *virtual edge* between $X_i$ and $X_j$ [39].

We provide a proof by contradiction. Suppose that $\hat{G}$ contains cycles. Suppose $C = (X_1, ..., X_c, X_1)$ is a cycle that does not contain any smaller cycles on its vertices. Since $G^*$ and $\hat{G}$ should have the same adjacencies (either via a real edge or a virtual edge), $G^*$ should also have edges in the location of all the edges of $C$.

- If $|C| > 3$, then the DAG has a v-structure, say, $X_{i-1} \rightarrow X_i \leftarrow X_{i+1}$. Therefore, there exists a subset of vertices $X_S$ such that $X_i \notin X_S$, conditioned on which $X_{i-1}$ and $X_{i+1}$ are independent. However, this conditional independence relation is not true in $\hat{G}$. This contradicts with quasi equivalence.
- If $|C| = 3$, then $G^*$ should also have a triangle on the corresponding three vertices, which contradicts the triangle condition.
- If $|C| = 2$, then suppose $C = (X_1, X_2, X_1)$. If none of the adjacencies in $\hat{G}$ to $C$ are in-going, then $C$ can be reduced to a single edge and the resulting directed graph is equivalent to $\hat{G}$ [16]. Hence, due to the sparsity penalty, such $C$ is not possible. If there exists an in-going edge, say from $X_p$ to one end of $C$, there will be a virtual or real edge to the other end of $C$ as well. Therefore, $X_p$, $X_1$, and $X_2$ are adjacent in $\hat{G}$ and hence in $G^*$, which contradicts the triangle condition. Also, if the edge between $X_p$ and one end of $C$ is a virtual edge, $X_p$ should have a real edge towards another cycle in $\hat{G}$, which, with the virtual edge, again forms a triangle, and hence contradicts the triangle condition.

Therefore, in all cases, quasi equivalence or the triangle assumption is violated, which is a contradiction. Therefore, $\hat{G}$ is a DAG.

**Proof of Theorem 2.**

From the first part of the proof, we obtained that $H(\hat{G}) \subseteq H(G^*)$. Therefore, by the contrapositive of part (b) in Assumption 2 we have $|E(\hat{G})| \geq |E(G^*)|$. Now, due to the sparsity penalty we have $|E(\hat{G})| \leq |E(G^*)|$. This concludes that $|E(\hat{G})| = |E(G^*)|$.

## C  Derivations of Maximum Likelihood Objectives

### C.1  General Linear Gaussian Model

Let $B$ be a weighted adjacency matrix representing a directed graph (possibly cyclic) over a set of random variables $X = (X_1, \ldots, X_d)$. The linear Gaussian directed graphical model is given by

$$X = B^\mathsf{T} X + N,$$

where $N = (N_1, \ldots, N_d)$ contains the exogenous noise variables that are jointly Gaussian and independent. The noise vector $N$ is characterized by the covariance matrix $\Omega = \mathrm{diag}(\sigma_1^2, \ldots, \sigma_d^2)$. Assuming that $I - B^\mathsf{T}$ is invertible, we rewrite the linear model as

$$X = (I - B^\mathsf{T})^{-1} N.$$

Since one can always center the data, without loss of generality, we assume that $N$, and thus $X$, are zero-mean. Therefore, we have $X \sim \mathcal{N}(0, \Sigma)$, where $\Sigma$ is the covariance matrix of the multivariate Gaussian distribution on $X$. We assume that $\Sigma$ is always invertible (i.e., the Lebesgue measure of noninvertible matrices is zero). The precision matrix $\Theta = \Sigma^{-1}$ of $X$ reads

$$\Theta = (I - B)\Omega^{-1}(I - B)^\mathsf{T}.$$

The log-density function of $X$ is then

$$\log p(x; B, \Omega) = -\frac{1}{2}\log\det\Sigma - \frac{1}{2}x^\mathsf{T}\Theta x - \frac{d}{2}\log 2\pi$$

$$= -\frac{1}{2}\log\det\Omega + \log|\det(I - B)| - \frac{1}{2}x^\mathsf{T}(I - B)\Omega^{-1}(I - B^\mathsf{T})x + \text{const}$$

$$= -\frac{1}{2}\sum_{i=1}^{d}\log\sigma_i^2 + \log|\det(I - B)| - \frac{1}{2}\sum_{i=1}^{d}\frac{\left(x_i - B_i^\mathsf{T}x\right)^2}{\sigma_i^2} + \text{const},$$

where $B_i \in \mathbb{R}^d$ denotes the $i$-th column vector of $B$.

Given i.i.d. samples $\mathbf{x} = \left\{x^{(k)}\right\}_{k=1}^{n}$ generated from the ground truth distribution, the average log-likelihood of $X$ is given by

$$L(B, \Omega; \mathbf{x}) = -\frac{1}{2}\sum_{i=1}^{d}\log\sigma_i^2 + \log|\det(I - B)| - \frac{1}{2n}\sum_{i=1}^{d}\sum_{k=1}^{n}\frac{\left(x_i^{(k)} - B_i^\mathsf{T}x^{(k)}\right)^2}{\sigma_i^2} + \text{const}.$$

To profile out the parameter $\Omega$, solving $\frac{\partial L}{\partial \sigma_i^2} = 0$ yields the estimate

$$\hat{\sigma}_i^2(B) = \frac{1}{n}\sum_{k=1}^{n}\left(x_i^{(k)} - B_i^\mathsf{T}x^{(k)}\right)^2$$

and profile likelihood

$$L\left(B, \hat{\Omega}(B); \mathbf{x}\right) = -\frac{1}{2}\sum_{i=1}^{d}\log\left(\sum_{k=1}^{n}\left(x_i^{(k)} - B_i^\mathsf{T}x^{(k)}\right)^2\right) + \log|\det(I - B)| + \text{const}.$$

The goal is therefore to find the weighted adjacency matrix $B$ that maximizes the profile likelihood function $L\left(B, \hat{\Omega}(B); \mathbf{x}\right)$, as also in Appendix C.2.

## C.2  Linear Gaussian Model Assuming Equal Noise Variances

If one further assumes that the noise variances are equal, i.e., $\sigma_1^2 = \cdots = \sigma_d^2 = \sigma^2$, following similar notations and derivation in Appendix C.1, the log-density function of $X$ becomes

$$\log p(x; B, \Omega) = -\frac{d}{2}\log\sigma^2 + \log|\det(I - B)| - \frac{1}{2\sigma^2}\sum_{i=1}^{d}\left(x_i - B_i^\mathsf{T}x\right)^2 + \text{const},$$

with average log-likelihood

$$L(B, \Omega; \mathbf{x}) = -\frac{d}{2}\log\sigma^2 + \log|\det(I - B)| - \frac{1}{2n\sigma^2}\sum_{i=1}^{d}\sum_{k=1}^{n}\left(x_i^{(k)} - B_i^\mathsf{T}x^{(k)}\right)^2 + \text{const}.$$

To profile out the parameter $\Omega$, solving $\frac{\partial L}{\partial \sigma^2} = 0$ yields the estimate

$$\hat{\sigma}^2(B) = \frac{1}{n}\sum_{i=1}^{d}\sum_{k=1}^{n}\left(x_i^{(k)} - B_i^\mathsf{T}x^{(k)}\right)^2$$

and profile likelihood

$$L\left(B, \hat{\Omega}(B); \mathbf{x}\right) = -\frac{d}{2}\log\left(\sum_{i=1}^{d}\sum_{k=1}^{n}\left(x_i^{(k)} - B_i^\mathsf{T}x^{(k)}\right)^2\right) + \log|\det(I - B)| + \text{const}.$$

## D  Proof of Lemma 1

First note that a weighted matrix $B$ represents a DAG if and only if there exists a permutation matrix $P$ such that $PBP^\mathsf{T}$ is strictly lower triangular. Thus, $I - PBP^\mathsf{T}$ is lower triangular with diagonal entries equal one, indicating that $\det(I - PBP^\mathsf{T}) = 1$. Since $P$ is orthogonal, we have

$$\det(I - B) = \det\left(P\left(I - B\right)P^\mathsf{T}\right) = \det(I - PBP^\mathsf{T}) = 1$$

and

$$\log|\det(I - B)| = 0.$$

# E  Proof of Proposition 1

The following setup is required for the proof of both parts (a) and (b).

The true weighted adjacency matrix and noise covariance matrix are, respectively,

$$B_0 = \begin{bmatrix} 0 & b_0 \\ 0 & 0 \end{bmatrix}, \Omega = \begin{bmatrix} \sigma^2 & 0 \\ 0 & \sigma^2 \end{bmatrix}, b_0 \neq 0.$$

Note that

$$I - B_0 = \begin{bmatrix} 1 & -b_0 \\ 0 & 1 \end{bmatrix}, (I - B_0)^{-1} = \begin{bmatrix} 1 & b_0 \\ 0 & 1 \end{bmatrix}.$$

In the asymptotic case, the covariance matrix of $X$ is

$$\Sigma = (I - B_0)^{-\mathsf{T}} \Omega (I - B_0)^{-1} = \sigma^2 \begin{bmatrix} 1 & b_0 \\ b_0 & b_0^2 + 1 \end{bmatrix}.$$

Let $B$ be an off-diagonal matrix defined as

$$B(b, c) = \begin{bmatrix} 0 & b \\ c & 0 \end{bmatrix}.$$

**Proof of part (a).**

Plugging $B(b, c)$ into the least squares objective yields

$$\ell(B; \Sigma) = \frac{1}{2} \operatorname{tr} \left( (I - B)^{\mathsf{T}} \Sigma (I - B) \right)$$
$$= \frac{1}{2} \left( (b - b_0)^2 + (b_0 c - 1)^2 + c^2 + 1 \right) \sigma^2.$$

The contour plot is visualized in Figure 3a. To find stationary points, we solve the following equations:

$$\frac{\partial \ell}{\partial b} = (b - b_0)\sigma^2 = 0 \qquad\qquad \implies b^* = b_0$$
$$\frac{\partial \ell}{\partial c} = b_0(b_0 c - 1)\sigma^2 + c\sigma^2 = 0 \quad \implies c^* = \frac{b_0}{b_0^2 + 1}.$$

Since function $\ell(B; \Sigma)$ is convex, the stationary point $B\left(b_0, \frac{b_0}{b_0^2+1}\right)$ is also the global minimizer. Thus, without a DAG constraint, the least squares objective $\ell(B; \Sigma)$ returns a cyclic graph $B\left(b_0, \frac{b_0}{b_0^2+1}\right)$. If one applies a hard DAG constraint to enforce choosing only one of $b^*$ and $c^*$, we have

$$\ell\left(B(0, c^*); \Sigma\right) = \frac{1}{2}\left(b_0^2 + 1 + \frac{1}{b_0^2 + 1}\right)\sigma^2 > \sigma^2 = \ell\left(B(b^*, 0); \Sigma\right),$$

where the inequality follows from the AM-GM inequality and $b_0 \neq 0$. Therefore, under a hard DAG constraint, $B(b_0, 0)$ is asymptotically the unique global minimizer of least squares objective $\ell(B; \Sigma)$.

**Proof of part (b).**

Plugging $B(b, c)$ into the likelihood-EV objective (2) yields (up to a constant addition)

$$\mathcal{L}_2(B; \Sigma) = \log\left( \operatorname{tr}\left((I - B)^{\mathsf{T}} \Sigma (I - B)\right) \right) - \log |\det(I - B)|$$
$$= \log\left((b - b_0)^2 + (b_0 c - 1)^2 + c^2 + 1\right) + \log \sigma^2 - \log|1 - bc|,$$

with contour plot visualized in Figure 3b. To find stationary points, we solve the following equations:

$$\frac{\partial \mathcal{L}_2}{\partial b} = \frac{2(b - b_0)}{(b - b_0)^2 + (b_0 c - 1)^2 + c^2 + 1} + \frac{c}{1 - bc} = 0$$
$$\frac{\partial \mathcal{L}_2}{\partial c} = \frac{2b_0(b_0 c - 1) + 2c}{(b - b_0)^2 + (b_0 c - 1)^2 + c^2 + 1} + \frac{b}{1 - bc} = 0.$$

(a) Least squares objective.  (b) Likelihood-EV objective.

Figure 3: The contour plot of different objectives in the bivariate case (with $b_0 = 1.5$ and $\sigma^2 = 1.0$). Lower is better. The black star corresponds to the ground truth DAG $B(b_0, 0)$, while the blue circle and green triangle indicate the (other) stationary point(s).

Further algebraic manipulations yield three stationary points and their respective objective values:

$$
\begin{cases}
b^* = b_0 \quad, c^* = 0 & \implies \mathcal{L}_2\big(B(b^*, c^*); \Sigma\big) = \log 2 + \log \sigma^2 \\
b^* = \frac{b_0^2 + 2}{b_0} , c^* = \frac{2}{b_0} & \implies \mathcal{L}_2\big(B(b^*, c^*); \Sigma\big) = \log 2 + \log \sigma^2 \\
b^* = -\frac{2}{b_0} \quad, c^* = \frac{2}{b_0} & \implies \mathcal{L}_2\big(B(b^*, c^*); \Sigma\big) = \log(b_0^2 + 2) + \log \sigma^2.
\end{cases}
$$

The second partial derivative test shows that $B(-\frac{2}{b_0}, \frac{2}{b_0})$ is a saddle point, while the other solutions $B(b_0, 0)$ and $B\big(\frac{b_0^2+2}{b_0}, \frac{2}{b_0}\big)$ are local minimizers, as also illustrated in Figure 3b. With DAG penalty, the cyclic solution $B\big(\frac{b_0^2+2}{b_0}, \frac{2}{b_0}\big)$ is penalized; thus, the acyclic solution $B(b_0, 0)$ becomes the unique global minimizer of the objective $\mathcal{L}_2(B; \Sigma)$. For $\ell_1$ penalty, we have

$$
\left\| B\left(\frac{b_0^2 + 2}{b_0}, \frac{2}{b_0}\right) \right\|_1 = \left| \frac{b_0^2 + 2}{b_0} \right| + \left| \frac{2}{b_0} \right| = \frac{b_0^2 + 4}{|b_0|} > |b_0| = \| B(b_0, 0) \|_1 .
$$

Hence, $\ell_1$ penalty encourages the desired solution $B(b_0, 0)$ and makes it asymptotically the unique global minimizer of the likelihood-EV objective $\mathcal{L}_2(B; \Sigma)$.

## F Optimization Procedure and Implementation Details

We restate the continuous unconstrained optimization problems here:

$$
\min_{B \in \mathbb{R}^{d \times d}} \quad \mathcal{S}_i(B; \mathbf{x}) = \mathcal{L}_i(B; \mathbf{x}) + \lambda_1 \|B\|_1 + \lambda_2 h(B),
$$

where $\mathcal{L}_i(B; \mathbf{x}), i = 1, 2$ are the likelihood-based objectives assuming nonequal and equal noise variances, respectively, $\|B\|_1$ is the $\ell_1$ penalty term defined element-wise, and $h(B) = \text{tr}\left(e^{B \circ B}\right) - d$ is the DAG penalty term. We always set the diagonal entries of $B$ to zero to avoid self-loops.

The optimization problems are solved using the first-order method Adam [23] implemented in `Tensorflow` [1] with GPU acceleration and automatic differentiation. In particular, we initialize the entries in $B$ to zero and optimize for $1 \times 10^5$ iterations with learning rate $1 \times 10^{-3}$. The number of iterations could be decreased by deploying a larger learning rate or proper early stopping criterion, which is left for future investigation. Note that all samples $\{x^{(k)}\}_{k=1}^n$ are used to estimate the gradient. If they cannot be loaded at once into the memory, we may use stochastic optimization method by sampling minibatches for gradient estimation. Our code has been made available at https://github.com/ignavier/golem.

Unless otherwise stated, we apply a thresholding step at $\omega = 0.3$ after the optimization ends, as in [54]. If the thresholded graph contains cycles, we remove edges iteratively starting from the lowest absolute weights, until a DAG is obtained (cf. Section 4.2).

In practice, one should use cross-validation to select the penalty coefficients. Here our focus is not to attain the best possible accuracy with the optimal hyperparameters, but rather to empirically validate the proposed method. Therefore, we simply pick small values for them which are found to work well: $\lambda_1 = 2 \times 10^{-3}$ and $\lambda_2 = 5.0$ for GOLEM-NV; $\lambda_1 = 2 \times 10^{-2}$ and $\lambda_2 = 5.0$ for GOLEM-EV.

# G  Supplementary Experiment Details

## G.1  Implementations of Baselines

The implementation details of the baselines are listed below:

- FGS: it is implemented through the `py-causal` package [41]. We use `cg-bic-score` as it gives better performance than the `sem-bic-score`.
- PC: we adopt the Conservative PC algorithm [36], implemented through the `py-causal` package [41] with Fisher Z test.
- DirectLiNGAM: its Python implementation is available at the GitHub repository `https://github.com/cdt15/lingam`.
- NOTEARS: we use the original DAG constraint with trace exponential function to be consistent with the implementation of GOLEM. We experiment with two variants with or without the $\ell_1$ penalty term, denoted as NOTEARS-L1 and NOTEARS, respectively. Regarding the choice of $\ell_1$ penalty coefficient, we find that the default choice $\lambda = 0.1$ in the author's code yields better performance than that of $\lambda = 0.5$ used in the paper. We therefore treat NOTEARS-L1 favorably by picking $\lambda$ to be $0.1$. Note that cycles may still exist after thresholding at $\omega = 0.3$; thus, a similar post-processing step described in Section 4.2 is taken to obtain DAGs. The code is available at the first author's GitHub repository `https://github.com/xunzheng/notears`.

In the experiments, we use default hyperparameters for these baselines unless otherwise stated.

## G.2  Experiment Setup

Our experiment setup is similar to [54]. We consider two different graph types:

- *Erdös–Rényi* (ER) graphs [13] are generated by adding edges independently with probability $\frac{2e}{d^2-d}$, where $e$ is the expected number of edges in the resulting graph. We simulate DAGs with $e$ equals $d$, $2d$, or $4d$, denoted by ER1, ER2, or ER4, respectively. We use an existing implementation through the `NetworkX` package [17].
- *Scale Free* (SF) graphs are simulated using the Barabási-Albert model [4], which is based on the preferential attachment process, with nodes being added sequentially. In particular, $k$ edges are added each time between the new node and existing nodes, where $k$ is equal to $1$, $2$, or $4$, denoted by SF1, SF2, or SF4, respectively. The random DAGs are generated using `python-igraph` package [12].

Based on the DAG sampled from one of these graph models, we assign edge weights sampled uniformly from $[-2, -0.5] \cup [0.5, 2]$ to construct the corresponding weighted adjacency matrix. The observational data $\mathbf{x}$ is then generated according to the linear DAG model (cf. Section 2.1) with different graph sizes and additive noise types:

- *Gaussian-EV* (equal variances): $N_i \sim \mathcal{N}(0, 1), i = 1, \ldots, d$.
- *Exponential*: $N_i \sim \text{Exp}(1), i = 1, \ldots, d$.
- *Gumbel*: $N_i \sim \text{Gumbel}(0, 1), i = 1, \ldots, d$.
- *Gaussian-NV* (nonequal variances): $N_i \sim \mathcal{N}(0, \sigma_i^2), i = 1, \ldots, d$, where $\sigma_i \sim \text{Unif}[1, 2]$.

The first three noise models are known to be identifiable in the linear case [34, 43]. Unless otherwise stated, we simulate $n = 1000$ samples for each of these settings.

### G.3 Metrics

We evaluate the estimated graphs using four different metrics:

- *Structural Hamming Distance* (SHD) indicates the number of edge additions, deletions, and reversals in order to transform the estimated graph into the ground truth DAG.
- *SHD-C* is similar to SHD. The difference is that both the estimated graph and ground truth are first mapped to their corresponding CPDAG before calculating the SHD. This metric evaluates the performance on recovering the Markov equivalence class. We use an implementation through the `CausalDiscoveryToolbox` package [22].
- *Structural Intervention Distance* (SID) was introduced by Peters and Bühlmann [35] in the context of causal inference. It counts the number of interventional distribution that will be falsely inferred if the estimated DAG is used to form the parent adjustment set.
- *True Positive Rate* (TPR) measures the proportion of actual positive edges that are correctly identified as such.

In our experiments, we normalize the first three metrics by dividing the number of nodes. All experiment results are averaged over 12 random simulations.

Since FGS and PC return a CPDAG instead of a DAG, the output may contain undirected edges. Therefore, when computing SHD and TPR, we treat them favorably by considering undirected edges as true positives if the true graph has a directed edge in place of the undirected one. Furthermore, SID operates on the notion of DAG; Peters and Bühlmann [35] thus proposed to report the lower and upper bounds of the SID score for CPDAG, e.g., output by FGS and PC. Here we do not report the bounds for these two methods as the computation may be too slow on large graphs.

# H Supplementary Experiment Results

## H.1 Role of Sparsity and DAG Constraints

This section provides additional results on the role of $\ell_1$ and DAG constraints for Section 5.1, as shown in Figures 4 and 5.

(a) ER1 graphs with 100 nodes.

(b) ER4 graphs with 100 nodes.

Figure 4: Results of different sample sizes in the identifiable cases (i.e., *Gaussian-EV*). Different variants are compared: GOLEM-EV (with $\ell_1$ and DAG penalty), GOLEM-EV-L1 (with only $\ell_1$ penalty), and GOLEM-EV-Plain (without any penalty term). Lower is better, except for TPR. All axes are visualized in log scale, except for TPR.

(a) ER1 graphs with 100 nodes.

(b) ER4 graphs with 100 nodes.

Figure 5: Results of different sample sizes in the nonidentifiable cases (i.e., *Gaussian-NV*). Different variants are compared: GOLEM-NV (with $\ell_1$ and DAG penalty), GOLEM-NV-L1 (with only $\ell_1$ penalty), and GOLEM-NV-Plain (without any penalty term). Lower is better, except for TPR. All axes are visualized in log scale, except for TPR.

## H.2 Numerical Results: Identifiable Cases

This section provides additional results in the identifiable cases (Section 5.2), as shown in Figure 6. For DirectLiNGAM, we report only its performance on the linear DAG model with *Exponential* and *Gumbel* noise, since its accuracy is much lower than the other methods on *Gaussian-EV* noise.

(a) Normalized SHD.

(b) Normalized SHD-C.

(c) Normalized SID.

(d) TPR.

Figure 6: Results in the identifiable cases with sample size $n = 1000$. Lower is better, except for TPR (lower right). Rows: ER$k$ or SF$k$ denotes ER or SF graphs with $kd$ edges on average, respectively. Columns: noise types.

## H.3 Numerical Results: Nonidentifiable Cases

This section provides additional results in the nonidentifiable cases (for Section 5.3), as shown in Figure 7.

(a) ER1 graphs.

(b) ER2 graphs.

(c) ER4 graphs.

Figure 7: Results in the nonidentifiable cases (i.e., *Gaussian-NV*) with sample size $n = 1000$. Lower is better, except for TPR. Experiments are conducted on graphs with different edge density, namely, ER1, ER2, and ER4 graphs.

## H.4 Scalability and Optimization Time

This section provides additional results for investigating the scalability of different methods (Section 5.4). The structure learning results and optimization time are reported in Figure 8.

Figure 8: Results of large ER2 graphs in the identifiable cases (i.e., *Gaussian-EV*). The sample size is $n = 5000$. Lower is better, except for TPR. Due to the long optimization time, we are only able to scale NOTEARS-L1 up to $1600$ nodes. The $x$-axes and optimization time are visualized in log scale.

## H.5 Sensitivity Analysis of Weight Scale

This section provides additional results on the sensitivity analysis to weight scaling for Section 5.5, as shown in Figure 9.

Figure 9: Results of different weight scales in the identifiable cases (i.e., *Gaussian-EV*). The sample size is $n = 1000$. Lower is better, except for TPR.