[Reviews · NeurIPS 2020]

Review 1

Summary and Contributions: The paper examines the sparsity and acyclicity constraints and their role in methods that cast structure learning as an optimization problem (e.g., the recent NOTEARS algorithm). The authors formulate a likelihood-based score function, and show that incorporating sparsity and acyclicity regularization can yield an effective algorithm. The resulting unconstrained optimization problem is much easier to solve. The key idea is to render the acyclicity constraint into a soft constraint during optimization and then to resolve remaining acyclicity problems in the final stage of the algorithm.

Strengths: The paper is well structured and the writing is relatively clear. The task (DAG structure learning) is well-studied and there are existing algorithms to which the new approach can be compared. The authors approach takes a previously impractical approach and makes it practical.

Weaknesses: Despite its frequent use in the literature, the assumption of linearity is unlikely to be satisfied in practice, and the authors should provide more justification for adopting this assumption. The authors call their acyclicity constraint “a regularization term”, which it really isn't. It is, as they call it earlier, a “soft constraint”. As is typical in the literature, the authors assume that the underlying data generating process can be represented as a DAG. This is a huge assumption and one that needs to be more frequently made explicit. PC has been substantially improved over the years since it was introduced. Specifically, varieties of PC such as RobustPC improve the accuracy of structure learning. The authors should state what variety of PC they used in their evaluation (and use one of the more effective versions). The authors use structural Hamming distance (SHD) and TPR as measures of accuracy of the structure learning method. However, recent work (Gentzel et al. 2019; Peters & Bühlmann 2015) has argued against using SHD and purely structural measures in general. The authors should strongly consider using alternative measures or at least justifying why, in this case, structural measures are adequate. Gentzel, A., Garant, D., & Jensen, D. (2019). The case for evaluating causal models using interventional measures and empirical data. In Advances in Neural Information Processing Systems (pp. 11722-11732). Peters, J., & Bühlmann, P. (2015). Structural intervention distance for evaluating causal graphs. Neural computation, 27(3), 771-799.

Correctness: Yes, overall the claims and methodology appear correct.

Clarity: The paper is clear and readable.

Relation to Prior Work: The paper positions itself clearly with respect to prior work.

Reproducibility: Yes

Additional Feedback: Overall, this is a good paper and should probably be accepted. It makes a previously impractical approach practical, and provides a new entrant in the effort to develop methods for accurate learning the causal structure of DAG-models.


Review 2

Summary and Contributions: The paper proposes a sparsity constrained or a DAG constraint based soft regularization of a likelihood based objective to recover a DAG model behind a linear SEM with Gaussian noise that is quasi- equivalent to the ground truth. This is in contrast to the recent NOTEARS paper where they use a least squares objective with a strict DAG constraint. Imposing a hard DAG constraint seems non trivial although a softer version could be imposed. However, softer DAG constraint with least squares objective does not have any theoretical guarantees as far as I am aware. The key result the authors show is that if the original DAG has a poly tree skeleton, then just a likelihood objective along with just the l_1 sparsity constraint recovers an acyclic DAG that is quasi equivalent to the original DAG assuming some technical conditions like g-faithfulness. When the poly tree skeleton assumption is violated, the MLE regularized by sparsity constraint could result in a cyclic graph but upon additional soft regularization involving a soft DAG constraint it can be chosen to pick a DAG that is quasi equivalent to the ground truth. This formally establishes that Soft regularization involving l_1 sparsity and the DAG constraint along with the MLE objective can recover DAGs that are quasi equivalent to the ground truth DAG under faithfulness conditions. Authors also demonstrate how a least squares based objective with l_1 regularization is not enough to recover the acyclic solution for a simple X1->X2 case with equal noise variances. The authors demonstrate experimentally the benefit of MLE objective with sparsity + soft DAG constraint over the NO tears counter parts for various non Gaussian exogenous noise variables in a linear SEM in synthetic simulations. In the supplement further comparisons are given for with other score based, constrained based algorithms like PC.

Strengths: Novelty is theoretical results showing that MLE objective with soft DAG and/or l_1 constraints recovers DAGs that are quasi equivalent to the ground truth and systematic synthetic demonstrations for linear SEM models. Scalability of these methods to thousands of nodes like the NO TEARS with better performance in terms of SHD and TPR is very notable.

Weaknesses: These are mild weaknesses. I view this work quite favorably due to the key point that the authors stress on which is the use of the MLE objective with soft l_1 and DAG constraints over the NO TEARS manner of using least squares objective. My concerns are as follows: a) The regularization hyperparmaters - The authors say that they use specific weights for regularization of the l_1 and DAG constraints in Section E.2. What motivated the authors choice ? However, when comparisons for NOTEARS are made, what is the regularization used there? Can the authors demonstrate that by better search of hyper parameters NOTEARS would not outperform GOLEM ? Like with all regularized methods, how important is the strength of regularization for GOLEM and given causal discovery is largely unsupervised, what is the authors suggestion of choosing the weights ? Although this is a bit of a serious concern, in light of the authors main theoretical contributions, I would like to not penalize them pending authors reply. Its better to sweep a bunch of regularization values and then plot the TPR and SHD metrics for both methods for a fixed set of samples. This plot would be very useful. One can fix the ratio of lamda_1 and lamda_2 to something. b) Lemma 1 says that the log det term in the MLE already provides for a DAG constraint. However, I did not find the proof in the supplement. Its better the authors provide a proof for that claim. c) Except for Sachs the empirical evaluation is based on synthetic linear SEMs. Is there a way to remedy this ? d) It seems that the asymptotic proofs heavily on the asymptotic behavior of only the MLE objective - So wondering if the regularization weights play any part in the proof? This has to be made clear in the proof. ##### - Update after the rebuttal I looked at the authors response, other reviewers concerns etc.. Looks like the authors addressed everyone's concerns satisfactorily. I will keep my score and vote of acceptance

Correctness: Upto my reading of the proofs, I find that the proofs seem correct although I have some questions regarding them. Please see weaknesses section.

Clarity: Yes. I have some clarifications I have sought. But largely the need for MLE objective is made very clear.

Relation to Prior Work: Yes. Upto my knowledge, relation to prior work on linear Gaussian SEMs is very well done.

Reproducibility: Yes

Additional Feedback:


Review 3

Summary and Contributions: The authors are proposing to learn DAGS using the maximum likelihood loss supplemented with a sparsity inducing penalty and the DAG promoting penalty introduced in NOTEARS. They compare the reconstruction performance of their loss function with NOTEARS in the linear Gaussian case and provide theoretical and experimental evidence for the effectiveness of MLE + penalty with respect to the square loss and constraints in NOTEARS.

Strengths: - The problem of learning DAGs (relatively) efficiently is clearly of interest and important for the Neurips community - Some effort has been invested into the presentation which make the paper easy to read which I definitely appreciate. - The work contains a rich collection of numerical experiments (in the supplementary material)

Weaknesses: Even though the paper reads well in its current form, I found during the first reading that the presentation of the conceptual messages (which are the most important part in my opinion) were overshadowed by the algorithmic comparison between GOLEM and NOTEARS. The paper is dense and it is not necessarily clear what the take-away messages of the paper are. - I would encourage the authors to elaborate more on the differences between the findings their findings that are related to algorithmic implementations and the findings that pertain to the asymptotic of loss functions. - I think it is very important to discuss in much more details (and maybe earlier in the paper) the relevance in your setting of transforming a constraint objective into a penalized objective. By duality there exists a penalized version of a loss that is equivalent to the constraint form of this loss. But your results seem to indicate that the penalty parameter asymptotically vanishes for the MLE when it may diverge for the square-loss. I also think that a paragraph concerning the limitations of the current study should be included in the conclusion of the paper.

Correctness: In my opinion the method and the numerical simulations are correct.

Clarity: I found the paper to have a coherent structure and that it reads well.

Relation to Prior Work: I believe that this paper clearly mentioned its position relative to previous works.

Reproducibility: Yes

Additional Feedback:


Review 4

Summary and Contributions: The paper proposes GOLEM, a "Gradient-based optimization of dag-regularized likelihood for learning linear dag Model". The paper follows the recent NOTEARS approach of casting the DAG structure learning problem (in the linear case) as a continuous optimization problem instead of a combinatorical search through the space of all possible DAGs. While NOTEARS applies a hard constraint on the weighted adjacency matrix that ensures a DAG but yields a constraint optimization problem, the innovation of GOLEM is to encourage DAG-ness by a soft constraint via a regularized maximum likelihood, yielding an unconstrained optimization problem, which is much more efficient to solve. A final DAG is then obtained by a postprocessing heuristic. GOLEM comes in two variants, one assuming equal noise variance (GOLEM-EV) and one that does not have the restriction (GOLEM-NV). The latter yields generally better results but is more difficult to optimize, with motivates GOLEM-EV the former as initialization procedure that helps avoiding local optima.

Strengths: The paper addresses a very relevant topic, that is, learning a DAG model a.k.a./ Bayesian network from data. While it considers only the special case of a linear Gaussian network (disregarding categorical variables entirely), the setting is still relevant enough to be of practical interest. Learning an optimal DAG is an NP-hard problem, so some approximations/heuristics are inevitable to obtain some scalability. The empirical studies on synthetic data are extensive with different network sizes, edge distributions, and sample sizes. The results make a fairly strong case in favor of both GOLEM variants compared to NOTEARS in terms of graph recovery. On top of that, GOLEM-EV is also more scalable. There is no running time result for GOLEM-NV, though. Nevertheless, the overall picture is a clear advantage over the existing baseline, which is particularly relevant as the proposed learning approach comes with relatively little theoretical guarantees.

Weaknesses: The learned weighted adjacency matrix is only locally optimal due to a non-convex objective function and requires some heuristics to obtain a final DAG. However, empirical evidence suggests that the heuristic is working appropriately, which covers this weakness pretty well. In line with the NOTEARS paper, real data experiments are missing almost entirely. There is a small study on the (tiny) Sachs-network at the bottom of the appendix, but the results are very weak for all methods: even a trivial method that returns an empty graph would obtain SHD=17, better than GOLEM-EV and only slightly worse than GOLEM-NV and NOTEARS. I think it's fair to say that none of the methods is learning much on this data set. Adding more real-world studies would thus strengthen the paper substantially, as there is no guarantee that the synthetic networks are even close to optimal networks for real data sets. While doing that is not always easy due to the unavailability of a ground-truth graph, one could use a consensus-based evaluation approach or look at the predictive performance of the model.

Correctness: I did not spot any errors in the methodology and experiments. The claims about causality might be a bit inappropriate, though. The introduction correctly states the limited causal interpretation of a learned DAG (third paragraph). However, the "broader impact" section at the end disregards this notion entirely, and implies that the method learns "causal structures" regardless. Given that especially this section may be also read by non-experts who are unaware of the subtle differences in interpretation, it might be better to use a bit more cautious wording there.

Clarity: Quality of writing and the presentation of results are both good. See below for a few suggestions for further improvement.

Relation to Prior Work: Yes.

Reproducibility: Yes

Additional Feedback: Just a few minor comments regarding the presentation of the results: (1) I'm a bit confused about the use of SHD to compare DAGs (main paper), as opposed to CPDAGs (supplement). SHD is originally [Tsamardinos et al. Mach Learn 2006] defined on (C)PDAGs for a good reason, and I think also here it does make more sense since to look at the differences among equivalence classes rather than differences among DAGs. That said, numerical differences are relatively small. (2) When plotting SHD as a function of d, it might make sense to divide the SHD by the number of node pairs so that the quality of the learned graphs is comparable among different values of d. (3) Also plotting the x-axis on a logscale (at least distribution of ticks) would improve readability. (4) The papers refers to appendix E.x when referring to additional results, but the actual appendix section is F. ###################### Edit AUG 22: I read the other reviews and the author's response. It appropriately addresses the reviewer comments given the time and space constraints of a conference paper. I was already quite positive here and continue to vote for acceptance.

[Author Response · NeurIPS 2020]

We thank the reviewers for their insightful feedback. We are encouraged that they found our motivation and idea to
be novel (**R2**, **R4**), practical (**R1**), interesting (**R3**) and scalable (**R2**, **R4**). Moreover, we are grateful that reviewers
identified our contributions beyond just performance gains on different tasks (**R1**, **R2**, **R4**), and that **R2** appreciated our
theoretical contributions. Reviewers also found that our empirical studies are sound and convincing (**R2**, **R4**).

Below we first provide a recap on the goal of our work, and then give a point-by-point response to the comments.

**[Recap] What is our goal?** We investigated the role of sparsity and DAG constraints for learning linear DAGs,
and established the corresponding *asymptotic guarantees*. Inspired by that, we formulate a likelihood-based method
(GOLEM), and showed that one only has to apply soft sparsity and DAG constraints for structure learning.

@**R1, R4** – **Performance metrics:** Thank you for raising this issue. Following your suggestions, we have computed
and will report the normalized SHD and Structural Intervention Distance (SID) in the revision. We found that these new
metrics yield consistent observations with the existing metrics reported in the paper.

@**R2, R4** – **Real data experiment:** Thank you for the suggestion. We agree that these methods may not learn much on
the Sachs dataset; in fact, even nonparametric extensions of NOTEARS, e.g., GraN-DAG (Lachapelle et al., 2020), do
not perform well on this dataset, either. Given your comment, we have done and will include an additional experiment
based on SynTReN generator (Van den Bulcke, 2006), which simulates gene expression data with ground truth. For
each method, the SHDs, normalized SHDs and SIDs are: GOLEM-NV (35.9, 1.8, 128.4), GOLEM-EV (42.8, 2.1,
139.4) and NOTEARS-L1 (48.7, 2.4, 162.6), respectively. Note that lower is better for all three metrics reported here.

@**R1** – **Assumption of linearity:** Great point. Learning DAGs is challenging, and some assumptions (e.g., linearity)
are needed for developing estimation methods and corresponding theoretical guarantees, as in LiNGAM and NOTEARS.
We plan to extend our method to nonlinear cases as future work (L372-373).

@**R1** – **"Regularization" vs. "soft constraint":** Thank you for pointing this out. In order to avoid any possible
confusion, we will follow your suggestion and replace the term "regularization" with "soft constraint" in the final paper.

@**R1** – **Assumption of DAGs:** We agree that it is a huge assumption and will make it explicit in all relevant sections.

@**R1** – **Using a more effective variant of PC:** We experimented with the original PC and its variant CPC (Conservative
PC), and picked the latter which gave better results. We will include the detail in the revised paper. Thank you.

@**R2** – **Choice of regularization coefficients:** Thank you for the insightful comment. For NOTEARS-L1, we
experimented with several choices, and picked the one which yielded the best results (L570-573). For GOLEM, our
focus is not to attain the best possible accuracy with the optimal hyperparameters, but rather to empirically validate our
theoretical results. We thus did not perform a thorough hyperparameter search, and found that small coefficients suffice
to work well (L588-590). We will include more details on this.

@**R2** – **Proof of Lemma 1:** We will include the proof in the final version. Thank you.

@**R2** – **Regularization coefficients in asymptotic proof:** Thanks a lot for this suggestion, which helps improve our
presentation. We chose the sparsity term such that it scales with the order of $\mathcal{O}(\log n/n)$, as in the consistency result of
BIC score. We will clarify this in the revised paper before Theorem 1 (L153).

@**R3** – **"puts much emphasis on beating NOTEARS" and "comes with no little surprises for more optimal ML**
**loss is used rather than square loss from NOTEARS":** Thank you for the comment. We would like to respectfully
remind **R3** that our main contributions include establishing the *asymptotic guarantees* and showing that soft DAG
constraint is, under mild assumption, sufficient to recover the underlying DAGs (Section 3.1). The superior performance
of GOLEM came as a *by-product* of our theoretical results.

@**R3** – **"transforming DAG constraint into a penalty is such an advantage w.r.t. NOTEARS is questionable"**
**and "Is there a more efficient way for implementing constraint minimization than augmented Lagrangian":**
We would like to make it clear that our method *does not* involve augmented Lagrangian, since we proposed an easier
*unconstrained* problem with a soft DAG constraint term (L213-216). This is entirely different from the constrained
formulation of NOTEARS which increases the penalty to very large values, leading to optimization difficulties (L79-81).
We also provided asymptotic guarantees and extensive empirical results to justify our motivation.

@**R3** – **Impact of sparsity on NOTEARS:** It already includes a sparsity term, denoted as NOTEARS-L1 in Section 5.

@**R4** – **Causality and broader impact section:** Thanks for pointing this out. We will remove the claim of "causality".

@**R4** – **Use of SHD:** We use SHD to measure the discrepancy between DAGs, and SHD-CPDAG for equivalence
classes (L624-628), which is what **R4** nicely suggested. We will make these terminologies clear in the revised version.

We hope that the above discussion has addressed the reviewers' concern, and will incorporate all suggestions in the
final version. We once again appreciate the reviewers' time dedicated to reviewing our paper.

[Meta-Review · NeurIPS 2020]

This paper studies structure learning for DAGs and examines the role of sparsity and acyclic constraints in this problem. The paper shows that a regularized MLE objective recovers a DAG that is quasi-equivalent to the ground truth. The algorithm is also significantly more scalable than prior methods. This is a good step toward making structure learning in DAGs highly practical, and as such we recommend acceptance.